# Absorption of Nitrogen during Pulsed Wave L-PBF of 17-4 PH Steel

**DOI:** 10.3390/ma14030560

**Published:** 2021-01-25

**Authors:** Ben Brown, Joseph Newkirk, Frank Liou

**Affiliations:** 1Materials Engineering, Department of Energy’s Kansas City National Security Campus, Kansas City, MO 64147, USA; 2Material Science and Engineering Department, Missouri University of Science and Technology, Rolla, MO 65401, USA; jnewkirk@mst.edu; 3Mechanical and Aerospace Engineering Department, Missouri University of Science and Technology, Rolla, MO 65401, USA; liou@mst.edu

**Keywords:** additive manufacturing, 17-4 PH, nitrogen absorption

## Abstract

In the fabrication of 17-4 PH by laser powder bed fusion (L-PBF) the well-documented occurrence of large amounts of retained austenite can be attributed to an elevated concentration of nitrogen present in the material. While the effects of continuous wave (CW) laser processing on in-situ nitrogen absorption characteristics have been evaluated, power modulated pulsed wave (PW) laser processing effects have not. In this study the effects of PW L-PBF processing of 17-4 PH on nitrogen absorption, phase composition, and mechanical performance are explored using commercially available PW L-PBF equipment and compared to samples produced by CW L-PBF. PW L-PBF samples fabricated in cover gas conditions with varying amounts of nitrogen demonstrated reduced absorption levels compared to those produced by CW L-PBF with no effects on phase composition and minimal effects on mechanical performance.

## 1. Introduction

Laser powder bed fusion (L-PBF) produced components have garnered industrial acceptance for critical applications in recent years as a result of equipment advancements, improved performance characterization of the materials fabricated by this method, and a focus on design for additive manufacturing (AM) [1]. In the L-PBF process a laser heat source is used to melt a single layer of a component in a bed of free-flowing metal powder under an inert cover gas, typically pure argon or pure nitrogen. The fabrication process is repeated layer by layer from the bottom up until the component is completed. A wide variety of materials have been used to produce components by L-PBF, including numerous titaniums, aluminums, and steels [2,3]. Of the steel options, 17-4 PH is a popular selection for use with a variety of potential applications due to its high strength as a result of heat treatment [4,5].

With traditional wrought processing 17-4 PH is a primary martensitic structure with low amounts of retained austenite, generally below 10% [5,6,7]. Although 17-4 PH is well suited for production by L-PBF [8], when produced by this method austenite phase fractions of over 50% have been measured in the as-built structure [9,10,11,12,13,14,15]. This large amount of retained austenite leads to inconsistent heat treatment response and reduced strength compared to a traditionally processed martensitic structure [13,16,17]. The phenomenon of high retained austenite L-PBF 17-4 PH is well documented and can be primarily attributed to the absorption of austenite stabilizing nitrogen into the material from either the cover gas used in the process or the atomization gas used for powder production [10,12].

Nitrogen absorption from cover gas into a molten pool is not limited to L-PBF and has been explored in welding literature [18,19,20,21]. In laser welding, lasers with power greater than 1000 W are generally used compared to the 100 W to 1000 W range of lasers used in commercial L-PBF systems. Evaluation of nitrogen absorption from Yb: Fiber lasers used in welding shows that with high power lasers the plasma plume above the melt pool is sufficient to dissociate the diatomic nitrogen into monatomic nitrogen greatly increasing absorption potential [22]. At the lower laser powers of L-PBF plasma plume generation has also been observed [23] and the absorption of nitrogen cover gas has been measured [12]. While in-situ nitrogen absorption has been documented, in comparison more nitrogen has been shown to be absorbed in laser welding where higher laser powers are used [21]. While 17-4 PH is not designed to be high nitrogen steel due to nitrogen’s strong austenite stabilizing effect, there are many steels designed to take advantage of high nitrogen concentration [24]. In some applications, depending on the material this absorption can have positive effects and tends to provide greater toughness and lower porosity than welds without increased nitrogen [20,21,25]. Although demonstrated in welding literature, the use of cover gas to tune performance for L-PBF material has not been extensively researched.

In L-PBF systems, both continuous wave (CW) and pulsed wave (PW) laser emission have been used as the primary heating source [26]. Within PW systems, both Q-switching [27] and power modulation [28,29] are used to achieve pulsing. In Q-switched Nd: YAG lasers, very high peak powers in the megawatt range can be achieved over very short nanosecond pulses. In modern commercially available PW L-PBF systems, power modulated Yb: Fiber lasers are used to achieve pulsed exposure. PW lasers produce peak powers that are equal to their CW counterparts with pulse durations in the millisecond range [30]. As surveyed by Demir et al. [26] the duty cycles used by PW lasers can range from below 0.01 to above 0.90 for L-PBF applications. Both types of wave emission have advantages depending on the application. In PW emission, smaller melt pools with large temperature gradients are produced. In CW emission where the duty cycle of the laser is equal to 1.0, larger melt pools that have a slower cooling rate than PW melt pools are produced. The differences in heat input cycles result in differences in microstructure and, therefore, performance as well [31].

The effect of retained austenite in L-PBF produced 17-4 PH is well documented with the cause of this phenomenon primarily attributed to elevated nitrogen. The exploration of in-situ nitrogen cover gas absorption has been predominantly done with CW L-PBF equipment and PW L-PBF has not been explored. The intent of this paper is to assess the effect of process variables on the absorption of nitrogen in powder modulated PW L-PBF and present any effect on the structure and material performance compared to what has been documented in CW L-PBF.

## 2. Materials and Methods

Samples for this study were produced with a Renishaw AM250 (Wotton Under Edge, Gloucestershire, United Kingdom) and an EOS M280 (Krailling, Bavaria, Germany) L-PBF system. Cover gas absorption and mechanical samples were fabricated on the AM250 whereas only mechanical samples were fabricated on the M280. The AM250 used a 200 W 1070 nm wavelength Yb: Fiber laser that is operated in a power-modulated PW mode. The M280 used a 400 W 1070 nm wavelength Yb: Fiber laser operating in CW mode. The AM250 in PW mode allows for laser power, point distance, exposure time, and hatch spacing to be controlled, as depicted in Figure 1, to scan each layer of the samples. The CW laser on the M280 was controlled by laser power, scan velocity, and hatch spacing scan pattern inputs. On the AM250 a fixed laser power of 200 W and hatch spacing of 90 μm, point distance was ranged from 45 μm to 65 μm and the exposure time was ranged from 65 μs to 85 μs. These combinations result in an effective velocity range of 529 mm s^−1^ to 1000 mm s^−1^ calculated as the ratio of point distance to the exposure time. A summary of the nine pulsed parameter combinations used is listed in Table 1. For the CW exposure samples, EOS standard GP1 (17-4 PH) parameters were used and are listed as combination 10. A rotation angle of 67° was used between each layer and no border scans were applied. Volumetric energy density, also known as specific energy input as defined by Simchi and Pohl [32], was used and modified for Equation (1) to quantify these exposure parameter combinations. Here, laser power *P* in watts, layer thickness *h* in meters, hatch spacing *hs* in meters, and laser scan velocity *v* in meters per second are used. Equation (1) is a modification of the common volumetric energy density where the duty cycle *δ* ranging between 0.0 and 1.0 is added as a multiplier to account for the PW exposure parameters. For the CW parameters, *δ* is equal to 1.0. This combination of pulsed parameters on the AM250 provided a wide range of energy inputs from 29 J mm^−3^ to 63 J mm^−3^.
(1)$$E=\frac{P}{h\times hs\times v}\times \;\delta$$

The laser pulse power profile of the AM250 was measured using a Thor Labs DET36A silicon photo detector (Newton, NJ, USA) equipped with a notch filter to filter out laser wavelengths in the build chamber and recorded with a Keysight DSO-X 3012T oscilloscope (Santa Rosa, CA, USA). For these measurements, a defocused laser was scanned on an anodized aluminum plate with the photo detector located in the build chamber pointed at the build surface. A measured intensity profile for a 50 μs, 75 μs, and 100 μs pulse as performed on the AM250 is shown in Figure 2. For the AM250 there is a damped power response and approximately 20 μs of power drop off as the laser is realigning to the next exposure point. For the short 50 μs pulse this drop-off is approximately 40 μs in duration. These characteristics are present in all three measured pulse durations. The evaluation of these curves shows that with increasing pulse duration, the percentage of energy delivered to the powder bed, compared to a CW exposure of the same duration, increases. The duty cycle for these exposures was approximated by measuring the area under a single pulse curve and comparing it to an equivalent constant intensity over the same duration. The resulting duty cycles for the 50 μs, 75 μs, and 100 μs are 0.54, 0.73, and 0.86, respectively. From these 3 measured values, a linear best fit was used to produce Equation (2) which relates duty cycle *δ* to laser pulse exposure time *et*. The duty cycle values for combinations 1 through 9 were extrapolated using Equation (2).
(2)δ=0.0064×et+0.23

Build plate temperature control was set to 80 degrees Celsius for temperature stability between the various builds. 50 µm layers were used for all experimental builds on the AM250 and 40 µm layers were used on the M280 per standard EOS processing. For mixed gas experiments gas was supplied by high purity argon and nitrogen bottles manually mixed in the build chamber and monitored in-situ during builds with a Pfeiffer Vacuum GSD320 gas analysis system equipped with a Pfeiffer Vacuum QMG 220 mass spectrometer (Aßlar, Hesse, Germany). The build chamber of the AM250 was kept at a slight positive pressure of 15 mbar for sample fabrication. Gas mixtures were actively monitored and manually held within 5% of target values throughout each build and oxygen levels maintained at less than 100 PPM as monitored by the onboard AM250 dedicated chamber oxygen sensor. The nitrogen cover gas for M280 builds was supplied by the equipment’s onboard nitrogen generator.

Argon and nitrogen atomized 17-4 PH powder of 15–45 μm particle size distribution with composition summarized in Table 2 was used for all experiments. Inductively coupled plasma optical emission spectroscopy (ICP-OES), inert gas fusion, and combustion were used for all composition measurements. ICP-OES was used for all elements except oxygen, nitrogen, hydrogen, carbon, and sulfur. Inert gas fusion was used for oxygen, nitrogen, and hydrogen, while combustion was used for carbon and sulfur. Typical 17-4 PH composition as controlled by ASTM A564 is also listed in Table 2 for reference. 

Cover gas absorption samples were fabricated as 10 × 10 × 15 mm samples in an array with consistent 16 mm spacing. Tensile samples were produced by fabricating cylindrical rods 8 mm in diameter and 45 mm tall. Tensile samples were oriented in an array with consistent 20 mm spacing. Samples were machined to final geometry per ASTM E8 for round subsize specimen 4 as detailed in Figure 3.

Machined tensile samples were tested at a constant crosshead speed of 0.02 in/min until 2.0% strain where speed was then increased to 0.2 in/min through failure. X-ray diffraction (XRD) measurements were taken on the absorption samples using a Bruker D8 Advance diffractometer (Billerica, MA, USA) with measurements taken in 0.05 degree increments with 0.5 s exposures at each step. No filter was equipped resulting in both MoKα_1_ (λα_1_ = 0.70930 Å) and MoKα_2_ (λα_2_ = 0.71359 Å) wavelengths being used. Data was collected for these samples perpendicular to the build direction on the as-built outer surface.

## 3. Results and Discussion

### 3.1. Nitrogen Covergas Absorption

To evaluate the nitrogen absorption characteristics of PW L-PBF 17-4 PH a range of laser parameters were tested in various mixed argon and nitrogen gas environments. A combination of five varying cover gas mixes ranging from full nitrogen to full argon incremented at 30%, 50%, and 70% substitutions was used. Only pure nitrogen cover gas was used for the EOS M280 CW L-PBF builds. A total of five builds using argon atomized powder, each with a different gas mixture, were produced with a total of three replicates for each of the nine parameter combinations randomly located on the build plate. Only one replicate from the pure argon build environment was tested. The resulting nitrogen concentration of the fabricated samples is presented in Figure 4. Data is listed by their parameter combination energy density as listed previously in Table 1. Outlier data points in the 34 J mm^−3^ 100% nitrogen and 53 J mm^−3^ 50% nitrogen data sets led to large confidence interval ranges. Results show that with increasing concentration of nitrogen in the build chamber cover gas there is a corresponding increase in absorbed nitrogen in the sample. Likewise, but to a lesser extent, as the energy density of the parameter set increases as a result of longer pulse durations and shorter point distances, increased amounts of nitrogen were absorbed. With increasing energy density, the higher availability of nitrogen in the cover gas leads to more absorption than what is seen at lower energy densities. This corresponds with the trend of larger melt pools and increased plasma generation as laser power or energy density increases [23,34], both of which result in more gas absorption into the liquid metal.

Although an increase in nitrogen concentration was measured, the maximum amount achieved with this range of parameters is still significantly less than the starting nitrogen concentration of the nitrogen atomized powder which measured 0.184 wt.%. Over the energy density ranges tested a maximum increase in nitrogen concentration as a result of in-situ PW L-PBF absorption is observed at approximately 0.007 wt.% over base powder concentration. This amount is about half of that measured in CW L-PBF by Meredith et al. at 0.017 wt.% [12]. When comparing PW L-PBF to CW L-PBF, it’s been shown that consolidation occurs at lower energy and with smaller melt pools for a pulsed system [27]. Given a smaller melt pool and comparatively less plasma generation due to the lower energy density, it would be expected that less nitrogen is absorbed in general from a PW exposure than a CW exposure. For samples produced in pure argon, at energy densities above 44 J mm^−3^ as well as those produced at 42 J mm^−3^ and 43 J mm^−3^ a nitrogen reduction from what was measured in the starting powder was observed. Replicates of these samples were not tested in this study so the results are not definitive, but an apparent trend of actually reducing nitrogen concentration can be noted and warrant additional investigation as it follows documented behavior of nitrogen desorption of high-nitrogen steels in a pure argon environment during laser welding [35].

### 3.2. Equilibrium Solubility

In-situ nitrogen absorption quantities for both the PW exposure presented in this study and the CW exposure from literature are less than what can be found when comparing to concentrations resulting from nitrogen atomization. The composition of the 17-4 PH produced by nitrogen atomization as listed in Table 2 shows that with the increased contact time of the liquid material in the gas atomization process, more gas can be absorbed. Cooling rates for the L-PBF process are on the order of 10^6^ K s^−1^ [36] where the cooling rates typical in gas atomization are on the order of 10^4^ K s^−1^ [37]. To put the in-situ results into context, equilibrium solubility for nitrogen can be calculated for the two 17-4 PH compositions presented in Table 2. For this solubility calculation starting nitrogen in the liquid was set as 0 wt.% and was performed at 1600 degrees Celsius and 1 Atm gas pressure. Assuming ideal gas behavior the solubility of a gas into liquid metal can be calculated by Equations (3)–(7). Diatomic gasses such as nitrogen absorb into liquid metal as atoms as described by Sievert’s law resulting in the isothermal equilibrium constant *K* being proportional to the square root of the diatomic gasses’ partial pressure (Equations (3) and (4)). The free energy of the iron nitrogen solution $\Delta G_{rxn}$ is equal to the product of the ideal gas constant *R*, temperature of the reaction *T*, and the isothermal equilibrium constant *K* (Equation (5)). From here the activity coefficient of nitrogen in liquid iron *f_N_* can be calculated as the sum of the products of the constituent solute element concentrations (*j*) and their corresponding interaction coefficient for nitrogen $e_{N}^{j}$ (Equation (6)). Finally, the concentration of nitrogen in liquid alloy ${\left[N\right]}_{\left(wt\%\;in\;liquid\;alloy\right)}$ can be calculated per Equation (7). Free energy and constituent solute activity values listed in Table 3 as provided by Lupis were used [38] for the calculation.
(3)$$\frac{1}{2}N_{2\left(gas\right)}={\left[N\right]}_{\left\{wt.\%\;in\;liquid\;iron\right)}$$
(4)K= [N]{wt.% in liquid iron)(pN2)12
(5)$$\Delta G_{rxn}=3600+23.9\times T=-R\times T\times \ln K$$
(6)$$\log f_{N}=\;\sum e_{N}^{j}\left[j\right]$$
(7)$$\log K=\;\log f_{N}+\log{\left[N\right]}_{\left(wt.\%\;in\;liquid\;alloy\right)}$$

Results of the equilibrium solubility calculation are summarized in Table 4 corresponding to the two starting powder compositions. The measured nitrogen concentration for the nitrogen atomized powder lot was found to be 0.184 wt.% where the concentration of nitrogen in the argon atomized powder lot was 0.0219 wt.%. The small amount in the argon atomized lot can be attributed as a residual from the steel making process and not as a byproduct of powder production. For both 17-4 PH powder lot compositions, the equilibrium solubility is calculated to be approximately 0.2 wt.%. This value is greater than both the maximum experimentally measured PW L-PFB nitrogen concentration using the argon atomized powder and what was measured directly in the nitrogen atomized powder lot. This difference between the theoretical equilibrium solubility limit and what was measured experimentally implies a kinetic limitation in the system as this chemistry is thermodynamically capable of absorbing higher amounts of nitrogen than what was experimentally measured. As noted in the absorption results in the previous section, as more nitrogen becomes available, more is absorbed when the solubility limit has not yet been reached.

More nitrogen was absorbed in the slower gas atomization process compared to the faster PW L-PBF process, but still under the solubility limit. Although a different 17-4 PH composition, the experimentally measured nitrogen concentration from the CW L-PBF by Meredith et al. [12] is more than the PW L-LPBF, but less than the gas atomization amounts. This further reinforces that with a CW laser exposing at a duty cycle of 1.0 creating larger melt pools at higher temperatures with more plasma being generated, more nitrogen can be absorbed as a result of longer exposure of the liquid to the nitrogen environment with elevated amounts of monatomic nitrogen. Similarly, by decreasing the duty cycle and reducing melt pool liquid duration as well as the plasma plume, less nitrogen is absorbed. By manipulating laser parameters from very short pulses to long continuous exposure, absorption of nitrogen can be tuned to a desirable level.

### 3.3. Microstructure Analysis

A selection of XRD patterns from the produced PW L-PBF absorption samples as well as the single combination for a CW L-PBF sample produced from nitrogen atomized powder in a nitrogen environment can be found in Figure 5. Energy densities listed are consistent with the parameter combinations listed in Table 1. Experimental combinations of nitrogen concentration and energy densities not displayed exhibited the same behavior as what is presented in Figure 5 and are omitted for clarity. Samples from the pure argon, 50% argon + 50% nitrogen, and pure nitrogen build sets were tested. Out of each of the three sets, samples fabricated with energy densities of 29 J mm^−3^ and 63 J mm^−3^ corresponding to low and high energy densities from Table 1 are presented.

For each crystallographic plane, patterns exhibit a double peak due to lack of filtering resulting both Kα_1_ and Kα_2_ wavelengths being collected. Extracted phase fraction and lattice parameters from the XRD patterns are summarized in Table 5 using the larger Kα_1_ peaks. Measurement orientations perpendicular and parallel to the build direction were performed for these XRD measurements and it was found that measuring sample surfaces parallel to the build direction resulted in low sensitivity and no retained austenite was measured in the PW L-PBF samples. Measuring perpendicular to the build direction increased sensitivity resulting in the measured quantities presented in Table 5. When the γ-austenite peak intensities are compared to a γ-Fe reference (PDF 00-052-0513) no predominant preferred orientation is found. This detection dependence possibly indicates elongated retained austenite resulting in more exposed cross section in the orientation perpendicular to the build direction. With the already low amounts of retained austenite measured perpendicular to the build direction, reduced cross-section could drop any signal below the XRD detection floor. This however would need to be verified by a technique such as EBSD.

All measured PW L-PBF samples contained between 1.6% and 3.7% retained austenite with a lattice parameter range of 3.599Å to 3.608Å. The remaining phase fraction of these samples was the martensite phase with a lattice parameter range of 2.872Å to 2.878Å. In contrast the CW L-PBF sample with high nitrogen measured a phase fraction of 97.3% retained austenite at a smaller lattice parameter of 3.557Å. The smaller unit cell is reflected in the peak shift to higher angles for the CW L-PBF sample XRD pattern.

The results in Figure 5 and Table 5 show that while there is some increase in nitrogen as a result of the varying laser and cover gas conditions during PW L-PBF, the difference in absorbed nitrogen over the parameter ranges tested is insufficient to stabilize elevated amounts of the austenite phase at room temperature as seen in the CW L-PBF samples fabricated with a high starting amount of nitrogen. This result is consistent with Murr et al. [10] who showed that with an argon atomized powder, fabrication in pure nitrogen was not enough to result in stabilized austenite. Meredith et al. [12] presented similar results showing that with CW exposure higher amounts of nitrogen can be absorbed with standard processing parameters, however the absorbed amount is insufficient to stabilize additional austenite.

To compare PW and CW L-PBF microstructure samples were built in pure nitrogen and pure argon with argon atomized powder with parameter combination 4 and evaluated by optical microscopy. These samples with a martensitic structure were prepared with Fry’s reagent and can be found in Figure 6a through Figure 6d. Both microstructures consist of large columnar grains passing through several layers as indicated by the visible melt pool boundaries with no observable sub grain structure. There is no apparent structural differences between the two samples as indicated by the XRD results between the pure cover gases used. Both PW L-PBF structures are comparable with those presented in literature produced by CW L-PBF where there is a low amount of retained austenite [10].

The CW L-PBF sample built with nitrogen atomized powder in pure nitrogen with parameter combination 10. Due to the different structure the Fry’s reagent used on the previous samples did not reveal structure and this sample was electrolytically etched at 1.1 V with 70/30 nitric acid and is shown in Figure 6c,d. This sample shows a significant departure from the samples built with argon atomized powder. The microstructure consists predominantly of smaller grains with a fully cellular dendritic sub grain structure. This reflects the phase composition measured by XRD showing an austenite phase fraction of 97.3%. Although the addition of nitrogen during the atomization process is sufficient to induce a difference in structure, the nitrogen absorbed in-situ during PW L-PBF is insufficient to result in any major structural changes.

### 3.4. Mechanical Properties

Tensile samples were also tested to evaluate nitrogen effects at the concentrations seen in this study. PW L-PBF samples were fabricated with the same nominal parameter combination as the optical microscopy samples having an energy density of 52 J mm^−3^ in pure argon and pure nitrogen environments. This combination was listed as combination 4 in Table 1. These parameters were selected for the mechanical samples as they demonstrated a high density as determined by the optical micrographs without the excessive edge curling seen in the higher energy density combinations. A total of ten cylindrical rods were fabricated in each environment were selected for testing. CW L-PBF samples were produced with the same combination as what was used for the optical microscopy samples have an energy density of 61 J mm^−3^ as a more direct comparison to what is seen in literature. For the CW L-PBF samples a total of ten cylindrical rods were fabricated for testing.

Results of the mechanical testing is summarized in Table 6 with stress-strain curves for the PW L-PBF powder samples shown in Figure 7. In addition to the ductility spread that has been documented in L-PBF 17-4 PH as a result of porosity [39], several of the nitrogen produced samples contained larger manufacturing defects resulting in reduced ductility to below 5% strain at failure. These samples are included in Table 6 results contributing to the nitrogen cover gas data spread and in the stress-strain curves in Figure 7 as denoted by the dashed lines. As seen in the Figure 6 micrographs, some porosity was present in the parameter combinations used to fabricate these samples which is the primary contributor for the ductility spread in the data set. The step increase in stress at 2% strain is a result of the cross-head speed utilized to increase testing speed. It can be seen that there is a strain rate dependence in the material, however it is consistent across the sample sets.

Sufficient nitrogen was not absorbed to stabilize elevated amounts of austenite and the mechanical performance variation between fabrication in 100% argon and 100% nitrogen is minimal. The samples produced in nitrogen did exhibit a higher mean ultimate tensile strength of 864 MPa compared to the 821 MPa of the samples produced in argon. Although a difference between the two lots is seen, the fact that only ten samples per lot were tested needs to be taken into account. For a more definitive evaluation of the potential nitrogen strengthening effect, more samples need to be tested.

In comparison to both sets of PW mechanical samples, the fully austenitic CW samples exhibited much lower yield strength, slightly higher ultimate strength, and significantly higher ductility than the martensitic PW exposure samples. The differences between the PW and CW lots can be attributed to the difference in phase as a result of nitrogen levels from the starting powder and not directly the laser exposure type used. The difference in nitrogen between these three sets of mechanical samples is relatively small and within the solubility of a 17-4 PH chemistry demonstrating that significant mechanical performance differences.

## 4. Conclusions

In this study the nitrogen absorption characteristics and their effects during PW L-PBF 17-4 PH were presented and the following conclusions were reached.

Nitrogen absorption, in addition to what was present in the base material, was shown to be dependent on the concentration of nitrogen in the cover gas as well as the laser exposure parameters used to fabricate samples.PW L-PBF exhibited less nitrogen absorption than CW L-PBF. Over the ranges tested the absorbed nitrogen was insufficient to result in any significant structure changeIt was found that the retained austenite in L-PBF 17-4 PH has no predominant preferred texture, yet detection was dependent on samples orientation.The small amount of absorbed nitrogen in the PW L-PBF samples resulted in a slight increase in ultimate tensile strength that is significantly less dramatic than the effects of large amounts of retained austenite.The presented results suggest the ability to use cover to manipulate the performance of L-PBF produced components. By controlling laser parameters and cover gas composition, absorption can be tuned for a particular application, either minimizing it allowing for either nitrogen or argon to be used interchangeably or maximizing it to improve mechanical performance.

## Figures and Tables

**Figure 1 materials-14-00560-f001:**
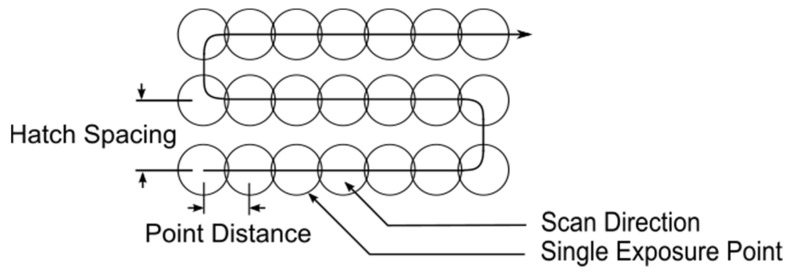
Diagram of Renishaw AM250 pulsing scan pattern [33].

**Figure 2 materials-14-00560-f002:**
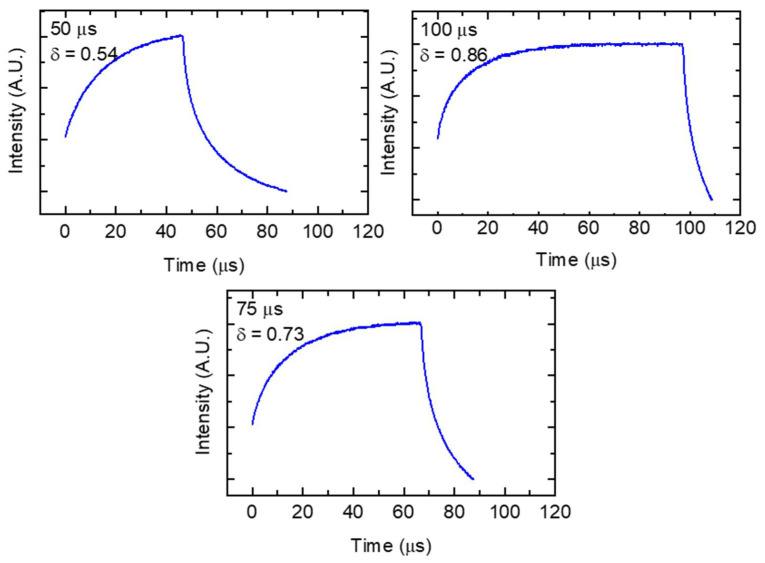
Measured laser intensity profile of a single pulse on the Renishaw AM250.

**Figure 3 materials-14-00560-f003:**
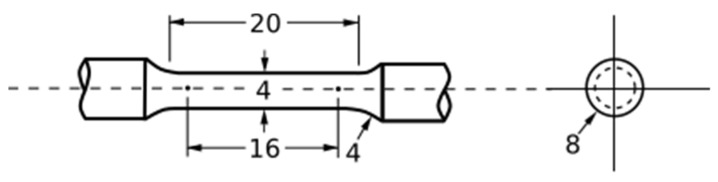
Machined tensile bar geometry used in mechanical testing in accordance with ASTM E8. All units in mm.

**Figure 4 materials-14-00560-f004:**
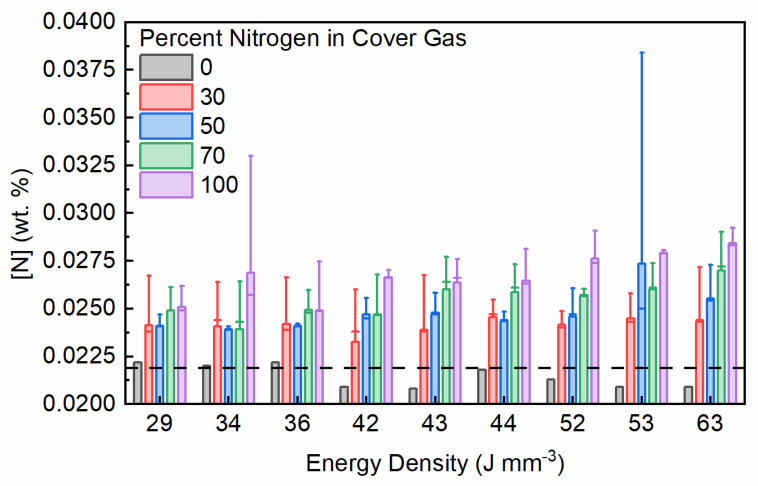
Calculated mean nitrogen concentration in PW L-PBF fabricated samples under mixed cover gas conditions with 95% confidence intervals indicated by error bars. Nitrogen concentration of 0.0219 wt.% in the argon atomized powder used to fabricate these samples is noted with a dashed line.

**Figure 5 materials-14-00560-f005:**
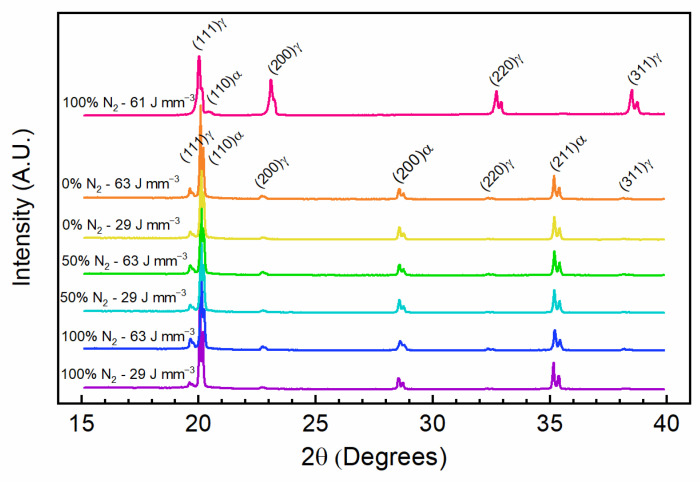
Mo source (λα_1_ = 0.70930 Å and λα_2_ = 0.71359 Å) XRD patterns of selected as-built PW and CW L-PBF samples. Data is labeled by percent of nitrogen in the cover gas and energy density per Table 1. Crystallographic planes are labeled with their associated phase (α-martensite, γ-austenite).

**Figure 6 materials-14-00560-f006:**
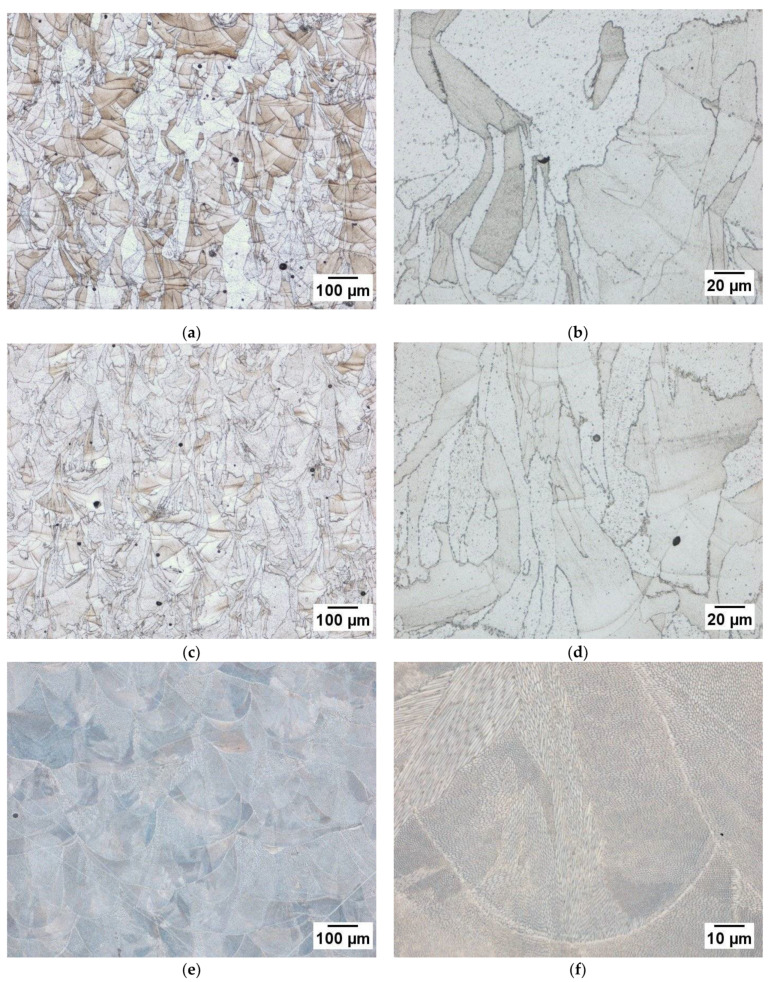
Optical microscopy images of samples fabricated from (**a**,**b**) argon atomized powder with PW L-PBF exposure in 100% argon; (**c**,**d**) argon atomized powder with PW L-PBF exposure in 100% nitrogen; (**e**,**f**) nitrogen atomized powder with CW L-PBF exposure in 100% nitrogen.

**Figure 7 materials-14-00560-f007:**
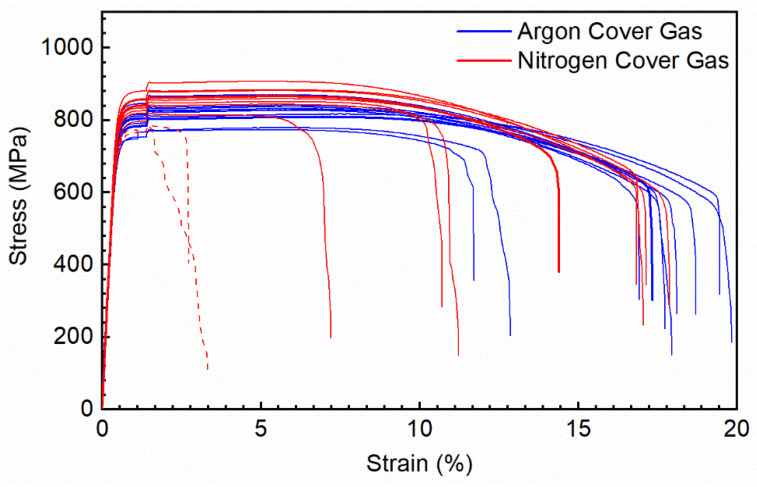
Stress-Strain curves of PW L-PBF fabricated tensile samples in both pure argon and pure nitrogen cover gas. Samples that exhibited manufacturing defects are denoted with dashed lines.

**Table 1 materials-14-00560-t001:** Summary of laser exposure settings used in experimental builds for both PW and CW exposures. Parameter combinations with a duty cycle < 1 indicate PW exposure where duty cycle = 1 indicates CW exposure.

Parameter Combination	Laser Power (W)	Hatch Spacing (μm)	Layer Thickness (μm)	Point Distance (μm)	Exposure Time (μs)	Scan Velocity (mm s^−1^)	Duty Cycle	Energy Density (J mm^−3^)
**1**	200	90	50	45	85	529	0.75	63
**2**	200	90	50	45	75	600	0.71	53
**3**	200	90	50	45	65	692	0.65	42
**4**	200	90	50	55	85	647	0.75	52
**5**	200	90	50	55	75	733	0.71	43
**6**	200	90	50	55	65	846	0.65	34
**7**	200	90	50	65	85	765	0.75	44
**8**	200	90	50	65	75	867	0.71	36
**9**	200	90	50	65	65	1000	0.65	29
**10**	195	100	40	~	~	800	1.0	61

**Table 2 materials-14-00560-t002:** Composition of powders used in experiments compared to the ASTM industry standard for 17-4 PH.

Element	ASTM A564 (wt.%)	Argon Atomized Powder (wt.%)	Nitrogen Atomized Powder (wt.%)
Cr	15.00–17.50	16.25	15.32
Ni	3.00–5.00	4.336	4.53
Cu	3.00–5.00	4.21	4.41
Mn	1.0 Max.	0.1968	0.81
Si	1.0 Max.	0.39	0.37
Nb	0.15–0.45	0.3	0.24
C	0.07 Max	0.0171	0.06
P	0.04 Max.	0.0117	0.012
S	0.03 Max.	0.00149	0.004
O	-	0.0422	0.038
N	-	0.0219	0.184
Co	-	0.0024	-
Mo	-	0.0068	0.094
V	-	0.05	0.038
W	-	0.001	-
Al	-	0.002	-
Fe	Bal.	Bal.	Bal.

**Table 3 materials-14-00560-t003:** Constituent solute activity interaction coefficient values for nitrogen in iron.

**Element**	**Interaction Coefficient**
Cr	−0.046
Ni	0.0063
Cu	0.009
Mn	−0.036
Si	0.047
Nb	−0.067
C	0.103
P	0.045
O	0.05
N	0
Mo	−0.011

**Table 4 materials-14-00560-t004:** Calculation results for equilibrium solubility of nitrogen in liquid 17-4 PH compositions as listed in Table 1 at 1600 degrees Celsius and 1 Atm gas pressure.

Powder Composition	Measured Nitrogen Concentration in Powder (wt.%)	Calculated Equilibrium Nitrogen Concentration (wt.%)
Nitrogen Atomized Powder	0.184	0.2032
Argon Atomized Powder	0.0219	0.2178

**Table 5 materials-14-00560-t005:** Extracted phase fraction and lattice parameter values from XRD patterns in Figure 5.

Energy Density (J mm^−3^)	Nitrogen in Cover Gas (%)	Powder Atomization Gas	Austenite Phase Fraction (%)	Martensite Phase Fraction (%)	Austenite Lattice Parameter (Å)	Martensite Lattice Parameter (Å)
61	100	N_2_	97.3	2.7	3.557	2.832
63	0	Ar	3.1	96.9	3.605	2.876
29	0	Ar	2.4	97.6	3.601	2.874
63	50	Ar	2.4	97.6	3.599	2.874
29	50	Ar	1.6	98.4	3.599	2.874
63	100	Ar	3.7	96.3	3.600	2.872
29	100	Ar	1.9	98.1	3.608	2.878

**Table 6 materials-14-00560-t006:** Summary of tensile results. Mean value of the data set with 95% confidence interval in brackets.

Laser Exposure Type	Powder Atomization Gas	L-PBF Cover Gas	Elastic Modulus (GPa)	0.2% YS (MPa)	UTS (MPa)	Strain at Failure (%)
PW	Ar	100% Ar	174.3 (165.1–183.49	777.3 (756.3–798.3)	821.3 (800.7–841.8)	17.3 (15.8–18.9)
PW	Ar	100% N_2_	181.6 (173.2–189.9)	815.7 (791.1–840.3)	864.2 (834.9–893.4)	14.1 (9.9–18.2)
CW	N_2_	100% N_2_	200.6 (188.8–212.4)	547 (533.4–561.2)	913 (911.0–916.1)	43.3 (42.6–44.1)

## Data Availability

The data presented in this study are available on request from the corresponding author.

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
