# Peer review of "Absorption of Nitrogen during Pulsed Wave L-PBF of 17-4 PH Steel"

_materials, 2021, doi:10.3390/ma14030560_

Round 1
Reviewer 1 Report
The authors studied absorption of Nitrogen during Pulsed Wave and continued wave laser powder bed fusion of 17-4 PH Steel. Below are my criticisms.
- There are number of small peaks observed in the XRD data. What are they? Such peaks are crucial in steel study. Please use the log scale to plot the XRD data and identify them.
- To have better understanding the mechanical properties of 17-4 PH Steel, microstructural study by EDSD is needed.
- The total scanning area of sample is identical either by Pulsed Wave LPD or Pulsed Wave LPD technique. Each material has its limited solid solubility. Although the one with larger melting pool can possess higher N element, its normalized value (Wt. %) should be same. Can you explain this phenomenon? What is the solid solubility of N in your materials? Please provide experimental prove on such hypothesis.
- Based on the stress-strain curve, it seems the reproducibility of your samples low. So their commercial value is limited.
Author Response
Thank you for your review. Reply to your detailed points are below:
1. There are number of small peaks observed in the XRD data. What are they? Such peaks are crucial in steel study. Please use the log scale to plot the XRD data and identify them.
In the XRD data, only peaks for austenite and martensite where measured. I believe you’re referring to the double peaks for each plane that are a result of both the Kα1 and Kα2 wavelengths being used. For more clarification detail was added to section 3.3 on page 8 reiterating this experimental detail. Figure 5 was also reformatted to better identify the peaks that were identified.
2. To have better understanding the mechanical properties of 17-4 PH Steel, microstructural study by EDSD is needed.
While EBSD would undoubtedly provide additional textural information on this material, the authors feel that none of the conclusions made in this manuscript would greatly benefit from this extra data. Per the point however, additional discussion was added to the XRD data and optical microscopy in section 3.3 to better inform the mechanical performance. Further evaluation on the texture and orientation of the retained austenite would be an interesting separate study building off the work presented here.
3. The total scanning area of sample is identical either by Pulsed Wave LPD or Pulsed Wave LPD technique. Each material has its limited solid solubility. Although the one with larger melting pool can possess higher N element, its normalized value (Wt. %) should be same. Can you explain this phenomenon? What is the solid solubility of N in your materials? Please provide experimental prove on such hypothesis.
This is an interesting point. From the data presented in the manuscript, it appears that the reasoning is best understood through the analysis of the data in Figure 4. As higher energy density is applied to the material the mobility of the nitrogen gas increases as additional plasma is generated and melt pool volume increases. Likewise, as more nitrogen is in the cover gas, the availability of nitrogen increases as well. The data in figure 4 shows that for each energy density, the absorbed nitrogen increases with the higher percent’s in the coverages and the spread of high to low increases as energy density increases. As a result given equivalent solubility and surface area, absorbed values increase as the mobility and availability of nitrogen increase. Additional discussion in sections 3.1 and 3.2 were added to better convey this point.
4. Based on the stress-strain curve, it seems the reproducibility of your samples low. So their commercial value is limited.
The presented mechanical data for the PW L-PBF samples did show spread, however this is fairly common in AM materials (see results presented in reference 39). Another reviewer made a similar comment in regards to the statistical analysis that was applied to this data and as a result 95% confidence intervals are now presented rather than 1 standard deviation. From this refined analysis it can be seen that while there is spread in the data, the predicted ranges for mechanical performance across all three data sets are relatively tight.
Reviewer 2 Report
In my opinion, the schedule of the research carried-out by the authors is interesting, but treated in a very narrow way and unreliably executed. The 17-4 PH steel is a martensitic steel, precipitation strengthened. When planning the research, the authors should have better known its behavior during the LPBF process, as well as positive and negative impact of dissolved nitrogen on this behavior.
The entire paper seems to be a test report with no attempt to interpret the unreliably presented results.
Detailed remarks:
- Line 134
The powder characteristics, including shape and surface condition of particles, influence porosity of the LBBF-ed tensile specimens, so the authors should compare both kinds of the powder (atomized in argon or in nitrogen) with regard to shape and surface quality of particles, preferably by showing SEM images.
- Lines 155-157
Dependence of the XRD measurement results of retained austenite fraction in LPBF-ed steel on orientation of the measurement plane in relation to the build direction can be related to directional solidification, resulting in crystallographic texture of austenite. This should be explained and the measurements should be performed on the xy and xz planes.
- Equation 3
The quantities occurring in this equation must be more clearly explained. It is completely unclear, why and how the equilibrium nitrogen concentrations in both kinds of the powder were calculated.
- Figure 5
The diffraction lines shown in the figure must be described. The authors should know that the first peaks from γ (111) and α (110) appear for the 2θ angles not smaller but larger than 40°. Diffraction lines obtained for both kinds of the powder would be also helpful. In its present form, Fig. 5 is useless, as well as the related comments.
- Table 5 and Figure 6
The fractions of retained austenite given in Table 5 are not consistent with the microstructure description, e.g. in Fig. 6c (line 276). Moreover, the images in Fig. 6 were taken at low magnifications, so it is almost impossible to distinguish martensite and austenite.
- Figure 7
The authors should explain the reason for the jumping increase of stress with no strain increase above the yield strength, occurring on each curve. Because of a small number and large scatter of the results, as well as small number of the specimens, not only standard deviations but confidence intervals should be also determined.
- Table 6
The results in Table 6 are inconsistent with the tensile curves in Fig. 6 and illegible.
Author Response
Thank you for your review. In response to the comment “The entire paper seems to be a test report with no attempt to interpret the unreliably presented results.” additional discussion was added throughout in order to increase the overall quality of the manuscript to the level worthy of journal publication. Reply to your detailed points are below:
- Line 134, The powder characteristics, including shape and surface condition of particles, influence porosity of the LBBF-ed tensile specimens, so the authors should compare both kinds of the powder (atomized in argon or in nitrogen) with regard to shape and surface quality of particles, preferably by showing SEM images.
Unfortunately powder samples were not retained after the builds for this type of analysis. As stated in section 2, given that this is off the shelf standard gas atomized 17-4 powder with known particle size distribution of 15 - 45 μm and chemistry for both lots, other researchers should be able to replicate these results.
- Lines 155-157, Dependence of the XRD measurement results of retained austenite fraction in LPBF-ed steel on orientation of the measurement plane in relation to the build direction can be related to directional solidification, resulting in crystallographic texture of austenite. This should be explained and the measurements should be performed on the xy and xz planes.
Additional clarification and discussion were added to section 3.3 to better describe the retained austenite sensitivity. When evaluating orientation dependent sensitivity, it was found that measuring parallel to the build direction resulted in no retained austenite being measured. As mentioned in your review, this is an indication of highly textured retained austenite in the direction of solidification.
- Equation 3, The quantities occurring in this equation must be more clearly explained. It is completely unclear, why and how the equilibrium nitrogen concentrations in both kinds of the powder were calculated.
A more comprehensive step through of the calculation was added to section 3.2 as well as equations 4-7 to allow other researches to replicated the steps used in this manuscript.
- Figure 5, The diffraction lines shown in the figure must be described. The authors should know that the first peaks from γ (111) and α (110) appear for the 2θ angles not smaller but larger than 40°. Diffraction lines obtained for both kinds of the powder would be also helpful. In its present form, Fig. 5 is useless, as well as the related comments.
Figure 5 has been updated to more clearly label the peaks and their associated planes. In regards to the position of the peaks, you are correct for copper source X-rays, however for this work lower wavelength molybdenum x-rays were used resulting in peak locations at lower angles. Additional detail was added to the Figure 5 description to clarify this point.
- Table 5 and Figure 6, The fractions of retained austenite given in Table 5 are not consistent with the microstructure description, e.g. in Fig. 6c (line 276). Moreover, the images in Fig. 6 were taken at low magnifications, so it is almost impossible to distinguish martensite and austenite.
Descriptions of Figure 6 were fixed as well as additional columns were added to Table 5 to more clearly match data sets to optical microscopy images. Figure 6d was added at higher magnification to better show the sub grain structure in the description of the material made from nitrogen atomized powder. High magnification images were not added for the argon atomized powder material as not additional structure is visible in the sub grain as is with the nitrogen atomized mateiral.
- Figure 7, The authors should explain the reason for the jumping increase of stress with no strain increase above the yield strength, occurring on each curve. Because of a small number and large scatter of the results, as well as small number of the specimens, not only standard deviations but confidence intervals should be also determined.
Description and reason for cross head speed increase resulting in the stress jump noted by the reviewer were added to section 3.4. 95% confidence intervals were also calculated and added to data throughout the manuscript as a replacement for the previous 1 standard deviation description.
- Table 6, The results in Table 6 are inconsistent with the tensile curves in Fig. 6 and illegible.
With the addition of the confidence intervals, the PW L-PBF data in table 6 is more clearly matched up with the curves in figure 7. The intent of Figure 7 is primarily to show that there is some ductility spread but that the 20 data points represented by the PW L-PBF samples are relatively tight and line up with what has been previously reported in literature (see results presented in reference 39). Additional description was also added in section 3.4 to help clarify interpretation of the data.
Reviewer 3 Report
Why author use both terms (sample, specimen).
Author Response
Thank you for the review. Per your comment,consistent use of 'samples' is now used throughout the manuscript except for the description of the ASTM tensile bar shape as that is explicitly called ‘specimen 4’,
Round 2
Reviewer 1 Report
The authors have addressed most of my criticisms, thus I would like to recommend for publication. Thanks.
Author Response
Thank you for the helpful review. Per your English language and style review the manuscript was spell checked with some slight grammar amendments added.
Reviewer 2 Report
Remark to Figure 6.
Microstructure of martensite (Fig. 6 a,b) and austenite cellular structure (Fig. 6d) could be best shown on SEM images or on light microscope images at properly big magnifications. The precondition is a well prepared and etched polished section.
Remark to the conclusion 4 (line 358).
The statement “Retained austenite in L-PBF 17-4 PH is highly textured..” is highly exaggerated and not supported by examination results. The conclusion could be supported by comparing relative intensities of reflexes I(200)γ/I(111)γ; I(220)γ/I(111)γ;... for the CW-LPBF specimen with their corresponding relative intensities of reflexes from the nitrogen-atomized powder sample with chaotic orientation, or with the data taken from a standard database.
Author Response
Thank you for your review. Reply to your detailed points are below:
1. Microstructure of martensite (Fig. 6 a,b) and austenite cellular structure (Fig. 6d) could be best shown on SEM images or on light microscope images at properly big magnifications. The precondition is a well prepared and etched polished section.
Per your review a new set of optical micrographs were taken and added for Figure 6a-f. Higher magnification images for the PW L-PBF show the lack of sub grain structure more clearly than the previous set of images. A stronger Fry’s reagents was also used in this set of images for a darker etch. A higher magnification image of the CW L-PBF sample was also added that more clearly shows the cellular dendritic structure within the melt pool.
2. The statement “Retained austenite in L-PBF 17-4 PH is highly textured..” is highly exaggerated and not supported by examination results. The conclusion could be supported by comparing relative intensities of reflexes I(200)γ/I(111)γ; I(220)γ/I(111)γ;... for the CW-LPBF specimen with their corresponding relative intensities of reflexes from the nitrogen-atomized powder sample with chaotic orientation, or with the data taken from a standard database.
Agreed that as previously stated the term ‘texture’ was a poor choice of words. Additional discussion and comparison for preferred orientation are added on page 9. The conclusion on page 12 was also reworded to better represent the presented results.